# Spatial Aspects of Gardens Drive Ranging in Urban Foxes (*Vulpes vulpes*): The Resource Dispersion Hypothesis Revisited

**DOI:** 10.3390/ani10071167

**Published:** 2020-07-09

**Authors:** Bryony A. Tolhurst, Rowenna J. Baker, Francesca Cagnacci, Dawn M. Scott

**Affiliations:** 1Ecology, Conservation and Zoonosis (ECZ) Research and Enterprise Group, Huxley Building, University of Brighton, Brighton BN2 4GJ, UK; row_baker@hotmail.com; 2Research and Innovation Centre, Fondazione Edmund Mach, Via Mach 1, San Michele all’Adige, 38010 Trentino, Italy; francesca.cagnacci@fmach.it; 3School of Life Sciences, Huxley Building, Keele University, Keele ST5 5BG, UK; d.scott@keele.ac.uk

**Keywords:** red fox, *Vulpes vulpes*, resource dispersion hypothesis (RDH), kernel density estimators (KDE), patch size, patch distribution, core areas, urban ecology, urban mammals

## Abstract

**Simple Summary:**

Red foxes are a well-established species of urban ecosystems in the UK and worldwide. Understanding how foxes use urban landscapes is important for urban biodiversity and disease management. The Resource Dispersion Hypothesis (RDH) proposes that home range size is smaller in larger, better quality habitats and increases as such habitats become patchier. Here, we tested the RDH for foxes in the city of Brighton and Hove, focusing on predominantly residential areas, as foxes are reported to prefer residential gardens to other urban habitats. We compared fox range size according to extent of garden land cover and varying garden size and configuration. We tracked 20 foxes (12 males and 8 females) with satellite collars recording at 15 min intervals for several weeks over four seasons to determine their range size and internal structure. We then measured garden size and configuration within fox ranges using Geographic Information Systems (GIS). We found that foxes had smaller ranges in areas containing few, large, well-connected gardens and larger ranges where numerous smaller gardens were crisscrossed by internal barriers (e.g., fences, walls) or fragmented by other habitats. Our findings confirm the RDH, showing that habitat dispersion can be used to predict fox movement in cities with relevance to urban planning.

**Abstract:**

Red foxes are a well-established species of urban ecosystems in the UK and worldwide. Understanding the spatial ecology of foxes in urban landscapes is important for enhancement of urban biodiversity and effective disease management. The Resource Dispersion Hypothesis (RDH) holds that territory (home range) size is linked to distribution and richness of habitat patches such that aggregation of rich resources should be negatively associated with range size. Here, we tested the RDH on a sample of 20 red foxes (*Vulpes vulpes)* in the city of Brighton and Hove. We focused on residential garden areas, as foxes were associated with these in previous studies. We equipped 12 male and 8 female foxes with GPS collars recording at 15 min intervals during discrete seasons over four years. We regressed fox core area size against garden size, number of garden patches, and edge density within and between patches as extracted from GIS in a series of bivariate linear mixed models. We found that foxes used smaller core areas where gardens were large and well-connected and larger core areas where numerous, smaller gardens were fragmented by internal barriers (e.g., fences, walls) or bisected by other habitats such as managed grassland or built-up areas. Our findings confirm the RDH and help to inform future urban planning for wildlife.

## 1. Introduction

Increasing global expansion of urban areas typically has negative effects on biodiversity [1,2], yet a few species appear to benefit from human activities in towns and cities [3]. Animals that fit this category are deemed “synurbic” and include red foxes (*Vulpes vulpes*) [4,5]. The presence of urban wild mammals has many positive effects on human wellbeing [6], although conflict can arise when these are abundant and in close proximity to humans [7]. Red foxes (hereafter “foxes”) are the most widespread and arguably the most abundant urban wild carnivore [4]. Negative impacts of fox activity on humans include nuisance via fouling, digging, or noise disturbance, very occasional attacks on pets or people [7,8], and risk of transmission of zoonotic or veterinary pathogens [9,10]. Foxes carry no significant zoonoses in the UK, although they are considered important hosts of lungworm (*Angiostrongylus vasorum*), causing angiostrongylosis in dogs, which is under-diagnosed and potentially fatal [10]. In parts of Europe and globally, fox populations carry rabies (*Lyssa virus* sp.) [11,12] and the cestode *Echinococcus multilocularis*, which causes human alveolar echinococcosis [13]. Attitudes of UK urban dwellers towards foxes are generally accepting but can be polarized [6], for example, immediately following sensationalist media attention [8]. Foxes do not use urban habitats uniformly [14,15,16,17], and an understanding of how foxes respond spatially to habitat heterogeneity in towns and cities is important for effective management. Urban foxes are excellent subjects for testing hypotheses on meso-carnivore space use due to their typically higher density relative to rural populations [18], territoriality, and habituation to humans [4]. In the South of England where most UK urban fox studies have been conducted, foxes occur at medium to high densities configured in small, contiguous or overlapping group territories (e.g., Oxford [19,20], Bristol [7,21], Brighton [22]).

The Resource Dispersion Hypothesis (RDH) [23] provides a framework for understanding the existence of sociality in facultatively cooperative or non-cooperative carnivore species in terms of configuration of resources. Foxes form social groups under certain conditions [7] and are therefore good models for testing the RDH. Under predictions derived from the RDH, size of fox social group territory and home ranges (where the latter overlap) should be linked to richness, number, and size of resource patches [23]. Fox range size has previously been shown to be negatively related to resource diversity at the landscape scale [24] and richness at the patch scale [25], but neither study was conducted in an urban area. Further, one ramification of the RDH is the concept of expansionism versus contractionism, where territory and associated range size are additionally affected by patch dispersion, i.e., spatio-temporal variation in resource occurrence [23]. For contractor species, smaller home ranges are expected where resources are rich, aggregated, and temporally stable. The presence of territorial behavior in urban foxes suggests that resources are relatively aggregated and that some or all are limiting [23]. For successful reproduction, foxes require food, mating opportunities, and secure dens and diurnal resting sites over the course of a typical temporal cycle [7,16,26]). 

Urban foxes are commonly associated with private residential gardens (e.g., Toronto, Canada [27]; Wrocław, Poland [26]; Vienna, Austria [17]; and Bristol, UK [16]), where they are usually free from persecution and encounter cover for denning and resting sites. They also exploit food provided by householders, deliberately or otherwise (e.g., [14,28]), and this provision appears to be increasing [7,29]. Conversely, foxes appear to select against the intensively managed short grassland typical of urban public green spaces (for example: playing fields, bowling lawns, cemeteries, and children’s playgrounds) [14,16,18] in addition to paved, “man-made” surfaces including roads, pavements, and buildings (e.g., [14,16]).

Therefore, of the habitats available to foxes, residential gardens appear to contain relatively rich resource patches that are worth defending. Spatial features of gardens such as size, number, and configuration would thus be expected to influence urban fox ranging behavior. Using the RDH framework, habitat (resource) patches can be quantified according to spatial scale such that, at the patch scale, a single garden separated from others by an edge (for example, a hedge or fence) comprises a discrete patch. At the landscape scale, patches comprise an area of contiguous gardens bordered by one or more alternative urban habitat types. The number of patches at each scale and the extent (density) of the edge represent patch dispersion. 

In the current study, we explicitly tested three hypotheses in the context of urban ecology and the RDH. Hypothesis (H)1 (upon the acceptance of which H2 and 3 were contingent) was that availability of gardens should disproportionately affect the size of the home range relative to other urban habitats. Hypothesis 2 then held that the degree of fragmentation of patches at the garden scale (i.e., garden edge density) should be positively linked to fox home range size. In addition, patch size (at both garden and landscape scales) should be negatively linked to fox home range size, because foxes need to use a larger area to access the resources required for reproduction where these resources are fragmented. H3 was a corollary of H2 and held that number of patches at both garden and landscape scales should be positively linked to range size, as the more numerous the patches, the greater the patch complementarity i.e., the likelihood that different key resources are available, in a heterogenous landscape. 

## 2. Materials and Methods

### 2.1. Study Area 

The study took place within the city of Brighton and Hove in East Sussex, United Kingdom (Latitude = 50.82253, Longitude = −0.137163 [WGS84]). We sampled six residential zones encompassing a range of housing densities to representatively sample the existing variation in extent of man-made surfaces in residential areas (Figure 1). Housing density was calculated using census data from the Office of National Statistics for postcodes (Ordnance Survey Code-Point Open) and categorized as low < 30, medium = 30–60, and high > 60 usual resident households per hectare. 

### 2.2. Field Methods

Live capture took place between April 2012 and September 2015. We expended equal effort in capturing foxes in each of the six zones and aimed to achieve a balanced sample of females and males across the four seasons. Seasons were categorized as winter (December–February); spring (March–May); summer (June–August); and autumn (September–November). Foxes were captured in baited cage traps and fitted with tracking collars under anesthesia. Additional biometric data were collected during assessment prior to anesthesia or from anesthetized animals, including weight in kg, sex, body condition (poor, fair, good, very good), and breeding status (breeding, non-breeding, bred this year) if apparent (Appendix A). Approximate age was quantified using date of capture, tooth wear, weight, and breeding status (Appendix A) as follows: subadult (SA) (6 to 12 months old, following e.g., [30] or adult (>12 months old); Table 2). Social status was inferred from weight, age, and breeding status and from direct observation where possible. Only foxes in body condition of fair or above, at least 10 months old, and weighing >6 kg were collared. For detailed trapping, handling, and collar drop-off methodology, see [22]. Two types of tracking collars were employed: 15 commercially available Global Positioning System (GPS) “Tellus collars” with Very High Frequency (VHF) and Global System for Mobiles (GSM) modules (manufactured by FollowIt, Lindesberg AB, Bandygatan 2, SE 711 34, Lindesberg, Sweden); and five “WildScope” proximity collars bearing GPS (GSM), VHF, and Wireless Sensor Network (WSN) units [31,32]. 

### 2.3. Data Handling and Home Range Analysis

All collars were programmed to record location data via GPS satellites every 10–15 min, downloaded in the form of Universal Trans Mercator (UTM) Northing and Easting coordinates. Fox relocation data were “cleaned” by removing all “trap associated” data points (data from the time of release until 12 h afterwards) to exclude locations representing abnormal behavior following capture and anesthesia. To determine whether home range metrics were appropriate for our data (i.e., whether foxes were resident over the sampling period), we implemented exploratory analysis in R v 3.4.0 (The R Foundation for Statistical Computing 2013, Vienna, Austria) using plug-in program rhr [33]. We first plotted an area observation (AO) curve for visual assessment of asymptote, i.e., the point at which the area used by an animal ceases to increase as successive locations are added [34]. Then, we investigated site fidelity (where the area an animal uses is smaller than if movements were random [35]) using a linearity index and the mean-squared distance from the center of activity, as compared for the observed data versus simulated random trajectories. We considered a fox to be resident if either home range asymptote or site fidelity were reached. As 90% of foxes tracked (n = 18) exhibited both an asymptotic home range and site fidelity, and the remaining 10% (n = 2) exhibited site fidelity only, we proceeded with home range analysis on data truncated at asymptote within seasons for all foxes.

We explored the use of various common home range estimators that describe the utilization distribution (UD)—a probability density distribution of animal relocations in two-dimensional space (see [36]). These included local convex hull (LoCoH) [37], Brownian bridge (BB) movement models [38], and kernel density estimation (KDE) [39]. Range size for our study animals was generally very small compared to travel speed, and therefore BB models were unsuitable for our data, whilst LoCoH methods tend to underestimate range size [40]. Point KDE is intuitive and easy to compute and thus remains the method of choice for determining the UD [36], allowing direct comparison with many other studies. We quantified home range in terms of core area, i.e., the area most intensively used [41], which we calculated using Seaman and Powell’s area-independent method of core area estimation [42] implemented in rhr, as this is more meaningful than arbitrary isopleths [33]. In this method, the core area is defined as the area within the home range where the probability of occurrence (i.e., of animal relocation) is greater than would be expected from uniform use. This is computed by plotting proportion of home range area used against relative frequency of use, the threshold for which is identified by an inflection point. We acknowledged the effect of spatial autocorrelation by repeating this procedure for different relocation frequencies, across which core area size remained stable. 

Appreciating the importance of appropriate selection of smoothing parameters for accurate delineation of core areas in KDE [36], we performed extensive evaluation of bandwidth (h) estimators for our data, including using the median of the average daily travel distances for each animal (h_mdt) following [43]. Least squares cross validation (LSCV) and plug-in-the-equation estimators consistently produced highly fragmented (under-smoothed) core areas, a tendency noted for LSCV (particularly for GPS data) by [36]. Contrastingly, bandwidth estimates using h_mdt produced much larger and over-smoothed core areas for some animals, skewing mean values. The reference bandwidth (h_ref) did not under or over smooth and was therefore used throughout. 

### 2.4. Landscape Metrics and GIS

We extracted the spatial variables relating to both gardens and other urban habitats from each of the three predominant land-use types in ArcGIS, calculated from Ordnance Survey’s Master Map (OSMM) topography and Open Greenspace datasets (Table 1). The spatial extent of each habitat type was validated using satellite imagery (Bing maps and Google Earth). We calculated the ratio of absolute garden area (ha) to combined area of two other urban habitat types—managed grassland and man-made surfaces (ha)—by dividing the first value by the second. We derived spatial variables that described the extent and the configuration of residential garden habitat patches at the level of individual gardens and at the landscape scale for each core area. These included: mean patch (garden) size (m^2^) and associated standard deviation (sd) and coefficient of variation (cv), number of patches (gardens), and edge density (ED in m/ha). Landscape scale variables were derived by eliminating the boundaries between adjacent garden patches using the dissolve tool in ArcGIS to make larger patches separated by other urban habitat types. At the garden patch scale, ED was calculated as the total length of internal boundary (fences, hedges, or walls between abutting gardens) per ha of total garden area as a measure of fragmentation between gardens. At the landscape patch scale, ED was calculated as the total length of boundaries with other urban habitat types per ha of total core area and was a measure of fragmentation of garden habitat by other urban habitat types. For both scales, ED excluded the edge generated by the border of core areas. 

### 2.5. Statistical Analysis

To investigate the effect of spatial variables on fox core area size, we generated a series of bivariate linear mixed models (LMMs) in R v. 3.6.1 (The R Foundation for Statistical Computing 2019, Vienna, Austria) using the package lme4. The response variable (core area size) was log-transformed to stabilize variances and allow parameter estimation based on the assumption that the data were generated from a Gaussian distribution. The response was then regressed separately against a series of fixed factors to test each of the three hypotheses selected from an initial set after accounting for multi-collinearity (see Appendix A for correlation coefficients and scatterplot matrix) and discounted removal of collinear variables via variance inflation factor or similar procedures so as to retain all variables of biological importance. The selected fixed factors were ratio of garden to combined other habitats (H1); ED (garden scale); mean garden size (ha) (H2); number of patches at garden scale (log-transformed to meet assumptions of linear regression); and number of patches at landscape scale (H3). Sex of fox (male/female), season (spring/summer/autumn/winter), year of study (2012–2015), mean housing density across core area (usually resident households ha^−1^), and mean housing density squared (transformed to meet assumptions of linear regression) were controlled for via inclusion in additional bivariate models. Fox individual ID was included in each model as a random factor.

### 2.6. Ethical Approval

Foxes were anaesthetized under UK Home Office license (PPL 7007429) under the guidance of a veterinary surgeon. Live capture, animal handling, and anesthesia were conducted according to a protocol approved by the Animal Welfare and Ethics Committee, University of Brighton. 

## 3. Results

We GPS-tracked a total of 20 foxes (8 vixens and 12 dog foxes) in four seasons over a period of four years across three housing density categories (Figure 1 and Table 2). Overall mean core area size (µ ± SD) was 15.7 ha (±14.1). However, one vixen (A9) ranged over an unusually large core area (Figure 1 and Table 2), and therefore calculations were additionally computed for the data excluding this outlier. Mean core area size for the remaining 19 foxes was 13.2 ha (±8.7) or 132,400 m^2^, and corresponding mean garden size was 0.012 ha (±0.01) or 126.3 m² (±102.1). The mean proportion of core areas covered by gardens was 0.43 (43% (±16)). Therefore, on average, a fox core area encompasses 451 gardens ((132,400/126.3) = 1048 × 0.43). If we take the average housing density from our study of 44.6 (±22.9) resident households per hectare, one fox core area equates to 589 households (44.6 × 13.2). This is comparable to 451 gardens, as some of the households constituted flats without gardens. The proportion of core area comprising man-made surfaces was similar to that covered by gardens at 0.456 (45.6% (±13.3)) but more than four times larger than that covered by managed grassland (8% ± 6.6) or complex habitats with cover (1.6% ± 0.8). 

Core area size was negatively correlated with the ratio of garden area to combined other habitats (lm, *F*_1,18_ = 7.075, *p* < 0.05; Appendix A for coefficients), supporting H1. Core area size was also negatively correlated with mean garden size (lm, *F*_1,18_ = 5.406, *p* < 0.05) and positively correlated with edge density (lm, *F*_1,18_ = 4.553, *p* < 0.05), thus supporting H2, and number of patches at both garden (lm, *F*_1,18_ = 69.25, *p* « 0.001) and landscape scales (lm, *F*_1,18_ = 40.28, *p* « 0.001), thus supporting H3 (Figure 2). A negative trend was observed between core area and patch size at the landscape scale, which approached significance (lm, *F*_1,18_ = 4.112, *p* = 0.058). All other control variables (sex, season, year, and housing density) were non-significant (see Appendix A for a full list of coefficients). The strongest predictor of core area was number of patches (garden), which accounted for 78.2% of the variation in core area, followed by number of patches (landscape), which accounted for 67.4% (Table 3). The best-fit model from Akaike Information Criterion (AIC) was number of gardens, but the greatest change in deviance was attributed to garden size (Table 3). Edge density was positively correlated with number of patches at both garden (lm, *F*_1,18_ = 24.29, *p* < 0.001) and landscape scale, although the latter correlation was weaker (lm, *F*_1,18_ = 10.89, *p* < 0.05); Appendix A). CV: cross validation; ED: edge density.

## 4. Discussion

We tested the Resource Dispersion Hypothesis (RDH) in the context of urban fox spatial ecology by investigating the relationship between fox core area size and residential garden patch size and dispersion. Our findings confirmed Hypothesis 1, as core areas were smaller where the ratio of gardens to all other habitats was highest. We inferred from this that gardens comprised the highest quality habitat relative to the other habitats available. We can also accept Hypotheses 2 and 3, as foxes ranged further where gardens were smaller, more fragmented (H2), and more numerous (H3). In addition, our findings indicate that fox ranging behavior is influenced not only by resource richness per se but also by uneven (patchy) resource distribution. This was due to the persistence of a strong positive association between core area size and number of patches at both the garden and the landscape scales, even after the variation attributed to fragmentation was accounted for. Fragmentation therefore negatively impacts habitat quality in some function of resource richness.

However, foxes must also range over more patches, regardless of whether these patches are fragmented, because different patches provide different resources. To confirm this corollary of H3, it is necessary to directly quantify resources and demonstrate that resource heterogeneity is linked to patch configuration, which we strongly recommend for future work. Resources in urban gardens include supplementary food provided by householders, ponds, shrubs or mature tree cover for diurnal rest sites, and spaces under sheds and other outside structures for dens personal observations, [14,16,44]. Fox group size and social connectivity have been linked to supplementary food provision in urban areas [14,45], but a link between resources and urban fox range size and structure has not, to our knowledge, been empirically demonstrated thus far. Natural food items are likely to be more numerous in larger gardens, consistent with species-area relationships [46] and concomitant with fox generalist life history. Larger gardens may also attract foxes because they can remain largely undetected by householders. The extent to which human disturbance matters to foxes is likely to vary with location and fox personality, as the behavioral phenotype of “tameness” is affected by complex interactions of genetic, social, and environmental factors [47]. Nonetheless, several studies in European cities cite human disturbance as a negative factor in fox habitat selection (e.g., [15,26]). Although the lack of serious zoonotic disease in the UK means that humans are generally tolerant of foxes, foxes generally alter their activity patterns and foraging behavior to keep humans at a minimum distance [4].

Although up to 78% of the variation in core area size was explained by number of patches at garden and landscape scales, differences between individual foxes contributed over a quarter (27.7%) of the variance. This indicates that fox-specific factors were influential in addition to environmental features. Intrinsic factors such as social organization and stage of lifecycle are important determinants of animal spatial ecology (e.g., [48]). Further, spatio-temporal overlap between conspecifics is affected by social network structure and stability in higher density fox populations [30] affecting movement within and between group territories. In our study, dominance status was reliably assigned in only three of 20 foxes, and we had no information on network connections c.f. [45], hence, we could not control for social factors in our statistical models. This represents a limitation of the study and future work should incorporate these within the theoretical framework from the outset. However, social structure has been shown to affect spatiotemporal overlap to a greater degree than spatial overlap [49]. Therefore, core areas of dominant and subordinate individuals in the same social group are likely to be similar in size and location, even if they are occupied at different times. An unbalanced sample of adults and subadults prevented incorporation of age in our analysis, and welfare issues preclude collaring juvenile or younger subadults. However, age is reported to be an important driver of space use [30], and further studies are recommended to determine whether age effects could account for the large proportion of fox-specific variation in core area size. Seasonal and inter-annual variation did not affect core area in the current study, which may in-part be explained by temporal stability of garden-based resources (e.g., supplementary food [28]).

If we compare spatial statistics with other areas to determine representativeness of our sample, we see that the mean proportion of fox core areas comprising gardens in our study (42.8% ± 15.1) was similar to that reported by [14] in the UK city of Bristol (43.5% ± SD 17%; n = 59) for a similar variable. In [14], non-winter overlapping individual home ranges were used as proxies for territories, and therefore the authors’ findings were comparable to ours. However, in contrast, they found no association between home range size and the proportional area of gardens. This may relate to the authors focusing solely on back gardens, which were identified as being optimal fox habitats in Bristol, whereas our data were derived from all garden types. Range size was also computed differently in [14] to our study, with minimum convex polygons (MCPs) derived from 95% of the locations. Our assessment of mean number of gardens encompassed by an individual fox core area is comparable to the national estimate of one urban fox social group per 648 houses, as derived from fox den surveys, householder questionnaires, and GIS models in 14 UK cities [50]. Although our study did not estimate size of combined social group home range, fox core areas within the same social group overlap [14] such that the number of houses encompassed by individual core areas would be expected to be only slightly lower than that of the social group overall. Therefore, our estimate of 451 gardens and 589 households per core area is plausibly supported by the literature.

The strong association found here between small fox ranges and a few, large, well-connected gardens is significant because fox social groups in urban areas are contiguous or overlapping where population density is high [14]. Small, contiguous home ranges are therefore indicative of high fox densities, and host density is a key parameter in predictive disease modelling. Finally, the current study illustrates the applicability of the RDH to urban landscapes within the context of an increasing “greening” imperative to understand the function of and enrich towns and cities for wildlife.

## 5. Conclusions

Our findings support the RDH in that fox space use was closely related to garden habitat patch dispersion, with implications for management of infectious disease, and urban biodiversity planning. Small fox ranges were associated with few, large, well-connected gardens. Further work is needed to identify the key resources in residential gardens and determine how foxes use them.

## Figures and Tables

**Figure 1 animals-10-01167-f001:**
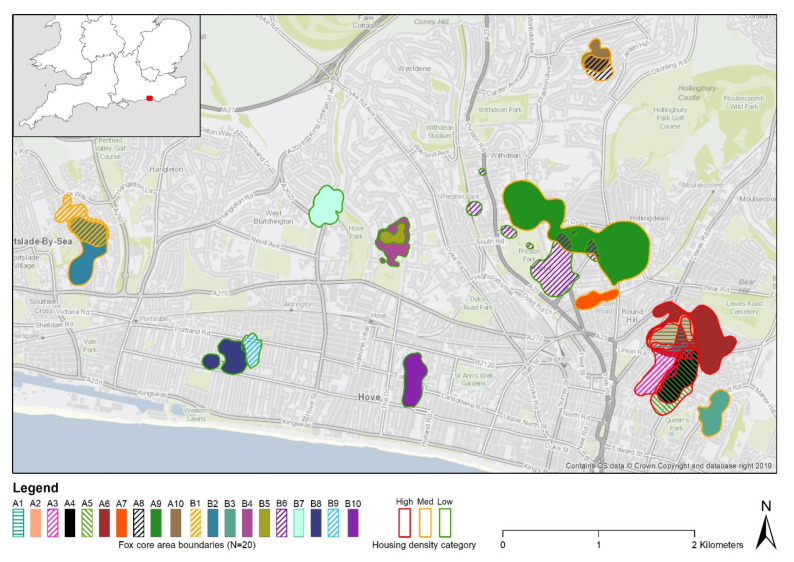
Location of fox core area ranges in six residential zones of the city of Brighton and Hove distributed across each of high, medium, and low housing density areas (low < 30; medium = 30–60; high > 60 usual resident households per hectare).

**Figure 2 animals-10-01167-f002:**
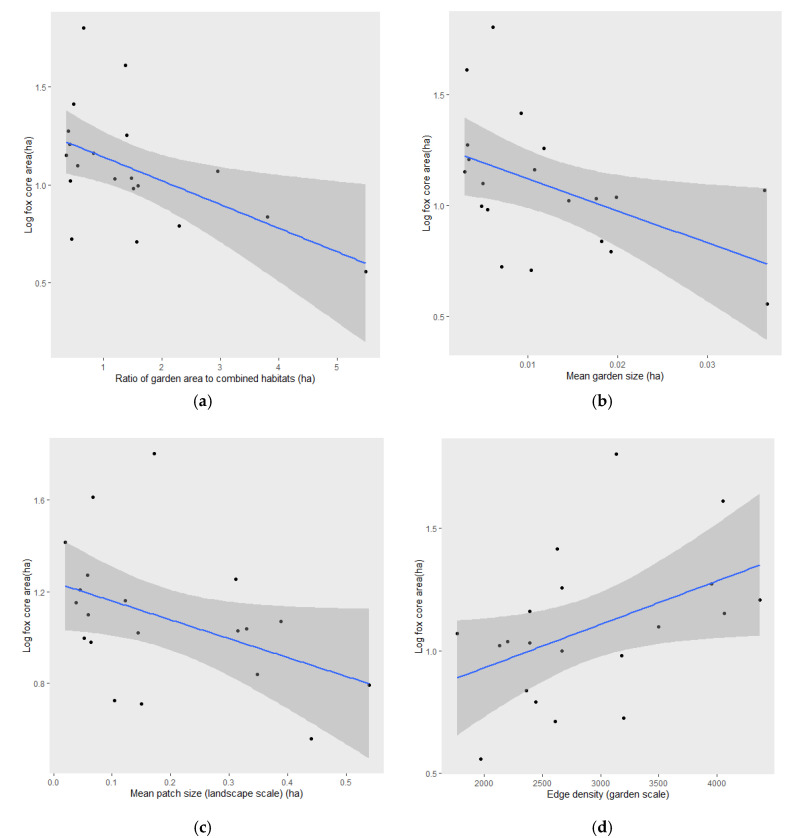
Predict plots for log-transformed core area size (ha) against: (**a**) absolute garden area (ha) as a ratio relative to all other habitats (H1); (**b**) mean garden size (ha), (**c**) mean patch size (landscape scale) (ha); (**d**) ED (garden scale) (H2); (**e**) number of gardens; (**f**) number of patches (landscape scale) (H3). Regression lines are shown in blue, confidence intervals in grey.

**Table 1 animals-10-01167-t001:** Land use types, attributes from Ordnance Survey Master Map (OSMM) used for selecting features and spatial statistics (habitats) derived from survey areas in Brighton and Hove for identifying landscape determinants of focal activity in urban foxes (*Vulpes vulpes*). ¹ Derived from Ordnance Survey Open Greenspace dataset.

Land Use Type	Spatial Statistic (Habitat)	Inclusions	Exclusions	OSMM Selected Attributes
1. Residential Gardens	Total garden area (ha) Patch size (m^2^) as mean, SD and CV (garden and landscape scale) Number of patches (garden and landscape scale) ED (m per ha) (garden and landscape scale)	Gardens of residential dwellings	Walled areas around flats Church yards School grounds	Feature Code = 1005 Descript_1 = Multi Surface
2. Public Green Space	Total area of managed grassland (ha)	Churchyards or burial grounds ^1^ Cemetery ^1^ Sports grounds ^1^ Public Park or Garden ^1^ Playing field ^1^ School grounds Extensive road verges/islands	Allotments ^1^ Hard ground fenced tennis courts	Feature Code = 1011 and make = natural + Feature Code = 1005 and make = natural
3. Man-made Surface	Total area of man-made surface (ha)	Manmade Surfaces: Roads and verges Car parks Tennis courts Hard surfaced playgrounds	Any natural environment within these	Make = manmade, *minus* theme = buildings and structures

**Table 2 animals-10-01167-t002:** Site and individual characteristics of 20 foxes GPS-tracked in Brighton and Hove in six residential zones with varying housing density, from April 2012 to September 2015. * Low < 30; medium = 30–60; high > 60 usual resident households per hectare. ++ truncated at asymptote except for * where asymptote not reached. ** Age and social status were estimated from a combination of month trapped, tooth wear, weight, body condition, and evident breeding status (see Appendix A for details). For A4, B4, and B5, additional behavioral observations were available to confirm social status (italics) [see 22]. A = adult (>10 months); SA = subadult (≤10 months).

Zone	Housing Density *	Season	Year	Fox ID	Sex	Social Status **	Age **	No. GPS Fixes ++	CA (ha)
1	High	Winter	2013	A1	M	Subordinate	SA	1491	12.54
1	High	Winter	2013	A2	F	Subordinate	SA	926	9.95
1	High	Winter	2013	A3	F	Subordinate	A	1052 *	16.10
1	High	Spring	2014	A4	F	*Dominant*	A	244	14.22
1	High	Spring	2014	A5	F	-	A	253	18.78
1	High	Spring	2013	A6	M	-	A	1339	40.90
2	Medium	Spring	2012	A7	M	Dominant	A	1619	5.13
2	Medium	Spring	2012	A8	M	Dominant	A	1750	6.19
2	Medium	Spring	2012	A9	F	Subordinate	SA	311	63.66
2	Medium	Autumn	2013	A10	M	Subordinate	SA	44	6.89
3	Medium	Spring	2013	B1	M	-	A	2802 *	14.51
3	Medium	Autumn	2015	B2	M	Dominant	A	267	18.02
4	Medium	Spring	2013	B3	M	-	A	2603	9.56
5	Low	Autumn	2013	B4	M	*Dominant*	A	805	11.73
5	Low	Autumn	2013	B5	M	*Subordinate*	SA	24	3.60
6	Low	Spring	2012	B6	F	Subordinate	A	1002	25.97
5	Low	Summer	2015	B7	M	Dominant	A	304	10.89
5	Low	Summer	2015	B8	M	Subordinate?	A	622	6.93
5	Low	Summer	2015	B9	F	Subordinate?	A	1605	5.13
5	Low	Summer	2015	B10	F	-	A	1063	10.51

**Table 3 animals-10-01167-t003:** Model fits for significant bivariate linear mixed models (lmm) explaining core area size in urban foxes. These were also run as generalized linear mixed models (glmm) to generate ∆ deviance (change in deviance attributed to the explanatory variable as a percentage of total deviance (residual + null)). * Akaike Information Criterion; ** percentage of the total variance explained by the random factor fox ID (i.e., residual variance/total variance).

Explanatory Variable	Adjusted *R*	AIC *	∆ Deviance (Residual/Total)	% Variance Random Factor (fox ID) **
Garden size	0.1882	8.4094	43.47	27.27
Number of patches (garden scale)	0.7822	−15.662	17.10	
Edge density (garden scale)	0.1575	9.1518	44.40	
Patch size (landscape scale)	0.1407	9.5468	44.87	
Number of patches (landscape scale)	0.6740	−9.8379	23.60	

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
