# Peer review of "Spatial Aspects of Gardens Drive Ranging in Urban Foxes (Vulpes vulpes): The Resource Dispersion Hypothesis Revisited"

_animals, 2020, doi:10.3390/ani10071167_

Round 1

Reviewer 1 Report

This is a commendable study for at least three reasons a) it presents an original and interesting data set on GPS tracked foxes, b) it sets the observations in a robust conceptual framework and c) it offers sensible interpretation and some comments of practical relevance. I recommend it should be published.

The paper offers a test of one prediction of RDH. Such field tests are notoriously difficult because of the technical problem of measuring directly the pattern in which (food) resources become available. It is therefore often necessary to resort to proxies. Insofar as the proxies are by definition approximate, the detection of biologically relevant relationships is difficult. Happily, in the case of urban/suburban foxes, there is a longstanding appreciation that large, often detatched, gardens represent rich food patches (having spent part of my childhood in the vicinity of the study area i can visualise it rather clearly). The analysis not only confirms this approximation, but examins its consequences in terms of the geometry of territorial configurations.

The paper draws on relevant literature. I will attempt to attach a PDF of a summary paper that the authors might find helpful. The data set has a lot of potential not explored in this paper (not in itself a criticism, but an opportunity). With 15 minute 'fixes' the detailed anatomy of home range use would withstand very detailed analysis with regard to the mechanics of how these data fit the range-size prediction of RDH. Further, and much more challenging, other predictions of RDH consider group size: I imagine the authors are already considering how their observations stack up against group-size predictions (a word of caution:the practicalities of testing these are considerable, and very often lead to the expression of poorly understood interpretation of fundamental RDH theory). Something perhaps for the future and for which the current paper will make a good foundation.

I appreciate the attempt to frame the findings in the context of conservation/management practicalities. While I would not quible with the arguments made, I think that in reality the application of these findings might need to be more nuanced - specifically, i am not sure fox density (territorial crowding) links in any simple way. My recollection, albeit from the early 70s, was that the very highest density of foxes (in the interface of rural suburbia) was associated with the humans who were least concerned and most pleased about the foxes' presence.

In short a good paper, the tip of a rich iceberg of data, with empirical and conceptual relevance, and likely to be of interest to a wide readership.

Author Response

Many thanks for your review. We have revised the paper extensively: please see attached MS.

Reviewer 2 Report

Research question

The question you tackle is interesting, particularly the link between core area size and residential gardens and the RDH. It therefore fills a gap in the already well filled urban fox literature.

Materials and Methods and discussion

Looking at study site and the core areas shown in figure 1, foxes caught have been probably socially (same family groups) and therefore in terms of space use linked.  I do not see the how this was accounted for in the analysis. Independence of data has therefore to be questioned. You could have checked that by integrating “zone” as a random effect as obviously many individuals have been cought at same zones. Also in the discussion this relation is not touched. You are additionally missing the information (at least it is not integrated in the manuscript) whether you are presenting data of subordinate or dominant individuals – also closely linked to the RDH and this should at least be discussed in the discussion section. Despite habitat use, you should discuss the social system of urban foxes and the consequences linked for your study and interpretation of data.

Author Response

Dear Reviewer 2

Many thanks for your comments, which we have replied to in full in the attached document.

The Authors

Reviewer 3 Report

Reviewers comments on Determinants of core area size in urban red foxes (Vulpes vulpes); testing predictions derived from the Resource Dispersion Hypothesis

This study combines GPS fixes with GIS within an urban landscape to derive core area use and likely predictors thereof.  The study design is good with an adequate sample size (n=20), appropriate analyses and reasonable interpretation of the results in addition to contextualisation of the study.  Weaknesses include:

  • The study attempts to make inferences about management and to frame parts of it within a conflict framework when in reality there is no data on conflict (e.g. surveys of residents) nor any data on negative impacts by foxes on people or people on foxes. Rather the study is more of a spatial ecology study of a small omnivorous predator in an urban matrix within the theoretical framework of RDH.  I would recommend that the authors focus on that aspect through the ms from the abstract to the discussion.
  • There is no measure of food availability in this study and yet the authors refer to large gardens as having more food and then cite two studies which both explored aspects of disease and foxes and neither of which systematically surveyed food availability in the urban matrix. The latter is a daunting task but remains a real challenge for most urban ecology spatial and diet studies.  You need more better references of the differences in food availability in different habitat types before you can make the inference that large gardens have more food.  In the absence of this and your own data on diet or food availability in your study area you need to restrict discussion to the spatial variables you measured and then suggest that this could be a function of more food and less barriers to movement BUT that you did not measure either and this needs to be done in future studies.
  • The study spans multiple years and it is not clear from the methods or results how year effects on the data have been assessed. In essence is May (spring) in 2012 the same as May in 2015?  Year should thus be included as a random term in the models.  Additionally it is not clear how long collars stay on animals relative to the season they have been categorised into.  Where data truncated if the animal was sampled in spring and summer months and if yes how did you decide which month/season to allocate the data too?
  • You mention reproductive status and condition was assessed but do not include them in the analyses as a covariate and I would have thought it would be important to ranging patterns with lactating females in particular likely to adjust core area size in accordance with increased costs and attendance. If you measure these variables then you should consider them in results as possible covariates of core size area.  In addition see note on use of two different collar types and whether these could influence data.
  • Please provide the collinearity matrix results (sup material?) and criteria for correlation being too high and whether you explored VIF.
  • You have not provided evidence of whether your data are spatially autocorrelated and how you deal with this. It does not suffice to state that foxes can move from one side of their core area to another inbetween GPS readings because in reality they do not.  In reality they are more likely to be closer to their previous GPS reading because readings are ‘fine scale’ and hence your data almost definitely are spatially autocorrelated.  You need to test this and if necessary remove it to avoid the problems it causes with interpretation of space use.
  • Please see specific comments in the edited pdf.

Author Response

Dear Reviewer

Many thanks for your detailed comments, which much improved the MS. Please find our response in the word doc attached.

The Authors

Reviewer 4 Report

Dear Authors,

This is an interesting study and provides more insight into spatial ecology of urban red foxes.

The text is written well. However, extension of some parts especially in the methods and discussion was requested. There are also suggested some minor corrections.

L.57 – Should be Lyssavirus spp. , additional citation is needed for the European distribution, see: Müller, T., Freuling, C. M., Wysocki, P., Roumiantzeff, M., Freney, J., Mettenleiter, T. C., & Vos, A. (2015). Terrestrial rabies control in the European Union: Historical achievements and challenges ahead. The Veterinary Journal, 203(1), 10-17. doi:10.1016/j.tvjl.2014.10.02

L. 57 -Delete microscopic - the cestode Echinococcus multilocularis is quite large; adult form can be up to 4.5 mm long, see https://www.cdc.gov/dpdx/echinococcosis/index.html

L. 62-64 – Provide citations

L. 125-126 and 202 – Being consistent the Authors should use small or capital letters for the season names

L. 145 – provide full name of rhr

Table 1 – use ha instead of Ha

Table 2 - - provide explanation for CA (ha) in the title of the table; is season associated with the capture of the animals or it is the only season when each individual was tracked? The total period of tracking of each individual should be provided in the table as well

L. 266 – isn’t conversion of 0.43 to 43% obvious? I would leave only 0.43, same L. 271

L. 271 – 4x larger –means 4 times larger? If yes please rephrase

Table 3 – Only the first four indicated rows in the table can be shaded, there rest can be left blank, this will also save ink for printing

L. 301 – I would write the authors instead of ‘they’

L. 303 – explain why KDEs and CA have advantage over MCP

Pages 9 – 10- As the Authors discuss Order Habitat Selection, the orders (2nd and 3rdorders) should be explained in the methods

Author Response

Please find our response in the word doc attached.
